# Point-of-Care Orthopedic Oncology Device Development

**Ioannis I. Mavrodontis \*, Ioannis G. Trikoupis, Vasileios A. Kontogeorgakos, Olga D. Savvidou and Panayiotis J. Papagelopoulos**

First Department of Orthopaedic Surgery, School of Medicine, National and Kapodistrian University of Athens, 11527 Athens, Greece; itrikoupis@med.uoa.gr (I.G.T.); vkonto@med.uoa.gr (V.A.K.); olgasavvid@med.uoa.gr (O.D.S.); pjportho@med.uoa.gr (P.J.P.)
\* Correspondence: imavrodontis@med.uoa.gr

**Abstract:** Background: The triad of 3D design, 3D printing, and xReality technologies is explored and exploited to collaboratively realize patient-specific products in a timely manner with an emphasis on designs with meta-(bio)materials. Methods: A case study on pelvic reconstruction after oncological resection (osteosarcoma) was selected and conducted to evaluate the applicability and performance of an inter-epistemic workflow and the feasibility and potential of 3D technologies for modeling, optimizing, and materializing individualized orthopedic devices at the point of care (PoC). Results: Image-based diagnosis and treatment at the PoC can be readily deployed to develop orthopedic devices for pre-operative planning, training, intra-operative navigation, and bone substitution. Conclusions: Inter-epistemic symbiosis between orthopedic surgeons and (bio)mechanical engineers at the PoC, fostered by appropriate quality management systems and end-to-end workflows under suitable scientifically amalgamated synergies, could maximize the potential benefits. However, increased awareness is recommended to explore and exploit the full potential of 3D technologies at the PoC to deliver medical devices with greater customization, innovation in design, cost-effectiveness, and high quality.

**Keywords:** individualized; patient-specific; endo-prostheses; instruments; point of care; 3D technologies; 3D printing; virtual reality; augmented reality

## 1. Introduction

In cases where no alternative options exist in cases such as bone reconstructions for limb salvage after oncologic resections, endo-prostheses must be tailored to the patient's unique anatomy and physiology. When and why artificial reconstructions of anatomical integrities are chosen depends on prognostic factors, technical considerations, short-to-long-term advantages and disadvantages, and other parameters [1]. Individualized implantable orthopedic parts have the potential to provide the following [2–4]:

1.  Superior mechanical performance tailored to the anatomical area free from stress-shielding phenomena leading to osteolysis.
2.  Superior biological performance that promotes bone ingrowth to achieve osseointegration.
3.  Optimal aesthetic outcomes with perfect fit.
4.  Optimal postoperative function with fast recovery and fewer complications and risks.

In contrast, compared with off-the-shelf counterparts, custom parts require a high level of expertise, careful handling, and quality control at all stages of their development, and they usually require higher costs and more time for pre-operative planning and development.

### 1.1. Enabling 3D Technologies

To facilitate the development of individualized medical devices at the point of care (PoC) and improve patient healthcare, it is necessary for certain emblematic 3D technologies to be utilized and appropriate synergies and workflows to be formed within scientifically

amalgamated and quality-controlled schemes. The term "3D technologies" refers to a broad gnostic field that includes the following elements, which enable the development of patient-specific solutions, including implantable medical devices (Figure 1):

1.  **3D medical imaging:** Tomographs (internal and external forms) and 3D scanners (external shapes) are used to capture high-fidelity anatomical image data for 3D visualization.
2.  **3D computer software:** These are used to (a) process medical image data; (b) plan pre-operatively and navigate intra-operatively; and (c) design, simulate, and optimize medical devices.
3.  **xReality:** These are virtual (VR), augmented (AR), and mixed (MR) devices used for (semi)immersive experiences to support pre-surgical planning, simulation, and surgical navigation. With VR (headsets), the user has the feeling of being part of an artificial world (virtual presence) that is different from actual physical reality, while with AR and MR (headsets, monitors, projectors), the user has a semi-immersive experience in which computer-generated content (text, images, animations) is superimposed over the user's actual environment.
4.  **3D printing:** This refers to (bio)material fabrication techniques for realizing physical medical devices from digital 3D models created with computer software.

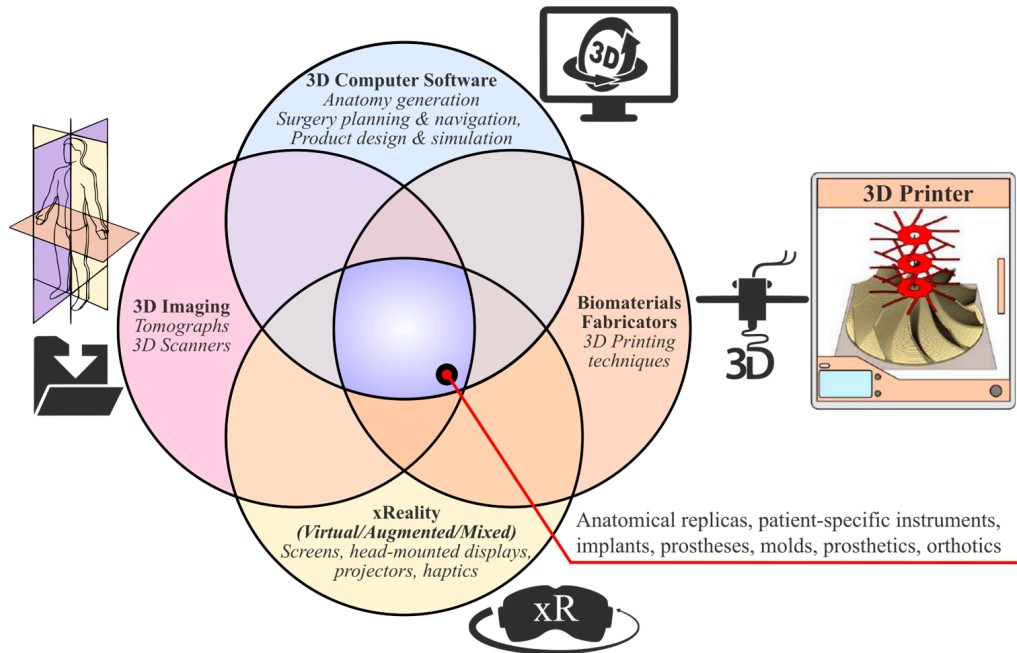

**Figure 1.** Three-dimensional technologies that enable the realization of individualized medical devices.

### 1.2. Three-Dimensional Printing of Individualized Medical Devices

According to ISO/ASTM [5], additive manufacturing (used synonymously with 3D printing) is defined as "a process of joining materials to make parts from 3D model data, usually layer upon layer, as opposed to subtractive manufacturing and formative manufacturing methodologies". There is a plethora of 3D printing techniques, each of which can be categorized into one of the seven classes: binder jetting, directed energy deposition, material extrusion, material jetting, powder-bed fusion, sheet lamination, and vat photo-polymerization.

Effectively, 3D printing is a tool-free method used for the primary shaping of complex freeform products. The 3D geometry emerges from the selective and successive layer-wise addition of material, with each layer corresponding to a cross-section of the product to be fabricated. This layer-by-layer principle enables almost unlimited and de novo design freedom to materialize extremely complex 3D geometries with intricate organic internal

and external features that vary in material density, geometry, and properties spatially across the part, per se. This way, manufacturing barriers are lifted, and the physical making of an object becomes easier than the digital 3D modeling of it. On the other hand, there can be technical flaws such as rough surfaces (the stair-stepping phenomenon), poor dimensional accuracy, anisotropic mechanical properties, micro-porosity, minimal feature size fabrication, and trapped powder particles (for powder-based techniques) [6].

Of the total value of 3D-printed products and services (USD 12.8 billion), the medical sector represents USD 1.65 billion [7]. Medical 3D printing has directly impacted more than 3 million patients and more than 5 million indirectly, with the largest growth expected in customized devices, surgical planning, and development and prototyping [8].

In medicine, intensive research is being conducted into 3D printing applications for making anatomical replicas, patient-specific instruments and molds, implants, (mega)endo-prostheses, orthotics and prosthetics, tissue-engineered constructs, and drug delivery products [9–11].

Numerous medically approved 3D printed devices have been implanted world-wide [12,13] with orthopedics and oncology being among the most common applications [14,15]. The process of 3D printing has been shown to provide beneficial solutions for bone substitution and limb salvage for oncological cases [16–18], and many break-throughs have been achieved owing to the utilization of custom 3D designs followed by 3D printing [14,19–22]:

1. Physical replicas aid in better anatomic anticipation and physical surgical simulations for pre-operative planning, communication, teaching, and training. Particularly useful is the utilization of multi-material 3D printing where hard and soft structures are fabricated to mimic biological tissues [23,24].

2. Patient-specific solutions help

    a. To improve intra-operative navigation and surgical (biomechanics and functions) and aesthetic results.
    b. To decrease operating time, surgical risks and errors, blood loss, radiation exposure/fluoroscopy shots, and postoperative complications and infections.
    c. To lower cost and development times for one-off unique parts compared with traditional manufacturing methods.
    d. To create better coordination and communication between multi-disciplinary and inter-epistemic teams without disrupting their workflows.

3. Biomimetic weight, surface, and topology optimized structures with favorable interconnected porous geometries can be implemented to fabricate scaffolds for tissue engineering and bone grafts [25] or for artificial bone endo-prostheses for increased osseointegration and bone-matching mechanical properties [26,27]. Thus, properties are topologically and selectively tuned along the monolithic part, per se.

4. Novel medical devices are accelerated and design and/or timely manufacturing at the PoC is empowered.

The 3D printing of medical devices at the PoC can be useful for making patient-specific medical devices, resulting in savings; however, apart from the necessary inter-epistemic and multi-disciplinary synergy formation, the main issues are economic and regulatory, and qualified personnel are needed [28–36].

### 1.3. xReality Applications

xR technologies offer alternative tools for decision making that can improve patient healthcare outcomes and economic value [37]. Researchers report that xR leads to improved education, training, and surgical skill acquisition, outperforming classic sessions and training approaches [38–40], while xR systems for intra-operative navigation and information visualization are promising in complex surgical procedures requiring high precision and accuracy such as oncology (tumor resection) and spine surgeries [41,42], as xR is more or

equally reliable and accurate and less time-consuming than ultrasound and fluoroscopy with less or equal anesthesia time and blood loss [43–47].

### 1.4. Scope

This paper aims to fill the gap in the literature by presenting and unfolding, step by step, an inter-epistemic workflow for developing individualized orthopedic oncological devices at the PoC harnessing the above-mentioned 3D technologies. For this purpose, a case study on pelvic endo-prosthesis development with a suitable synergy that is centered around such 3D technologies is presented, and the potential and challenges are reported.

## 2. Materials and Methods

A systematic approach to engineering design [48] was tweaked and blended with other approaches to create a 15-step inter-epistemic end-to-end workflow for the development of patient-specific medical devices at the PoC (Figure 2). To fully explore and exploit the capabilities of 3D printing, design with 3D printing (Dw3DP) [49] was harnessed as a toolbox to expand conceptualization and boost creativity toward new design spaces. To overcome the associated issues of 3D printing and to maximize product performance through the synthesis of geometrical and material compositions, design for 3D printing (Df3DP) principles were considered [50,51]. Df3DP and Df3DP were inserted into a divergence–convergence model similar to the Double Diamond design process [52]. To support mimicking natural forms and functions, the Biomimicry Institute's design spiral and taxonomy of biological strategies were employed [53].

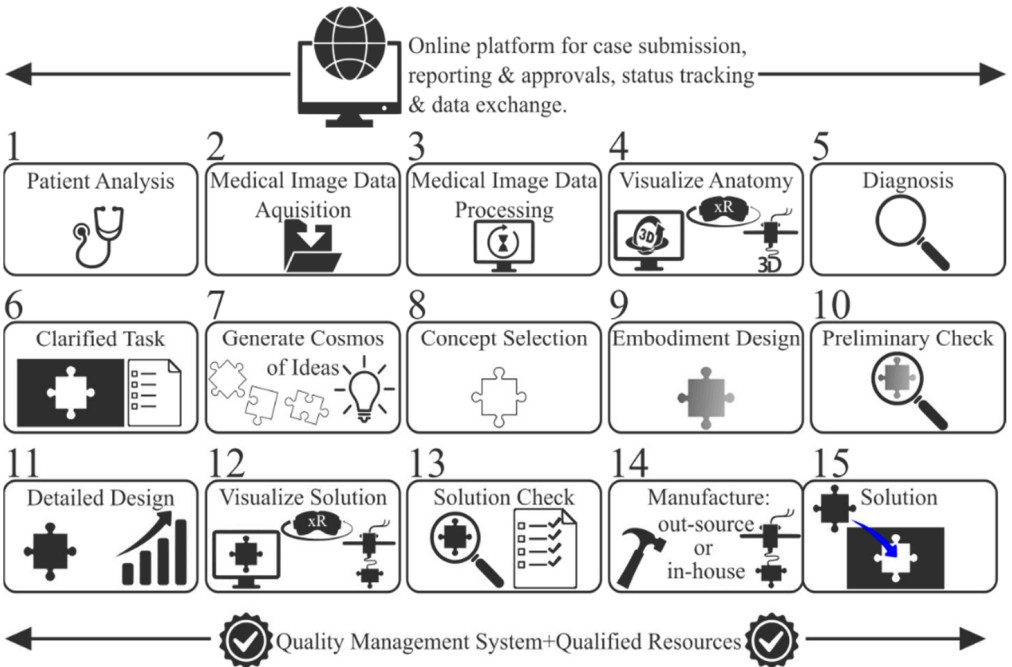

**Figure 2.** Inter-epistemic workflow for developing individualized orthopedic products.

Utilizing the aforementioned workflow, a case study of artificial pelvic reconstruction was carried out locally at a small 30-square-meter pilot PoC unit equipped with the hardware shown in Figure 3. In addition, an online platform was set up for status tracking and data exchange between the surgeon and the pilot PoC unit, supporting all phases described below.

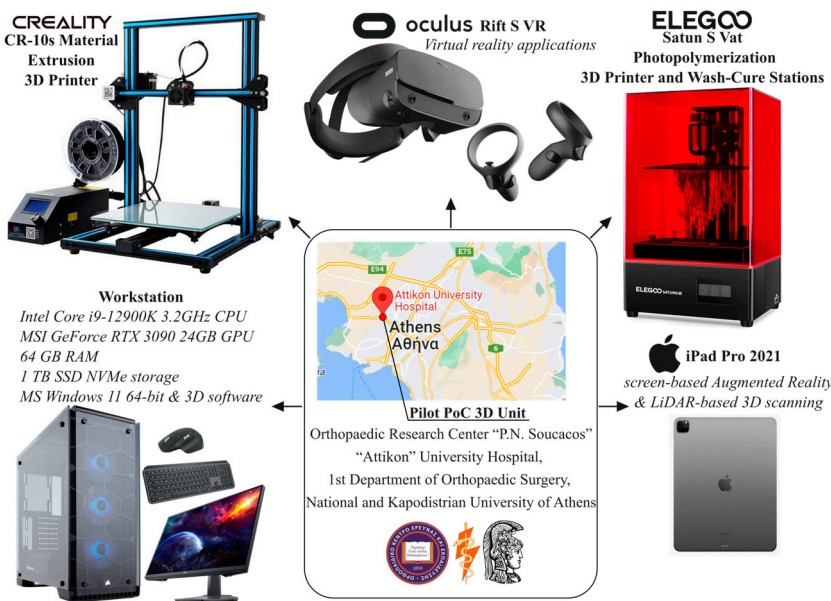

**Figure 3.** PoC equipment.

### *2.1. Step 1—Patient Analysis*

An adult male patient was diagnosed with pelvic osteosarcoma, and a two-step reconstruction approach was planned as per the prognostic factors and parameters and the tumor stage and site based on CT and MRI imaging (Figure 4). The ilium and sacroiliac joint were resected freehand en bloc (Type I plus partial IV plus partial II according to Enneking's classification [54]), and artificial reconstruction of the native bone structure with a non-bulky endo-prosthesis accompanied by patient-specific instruments was conducted.

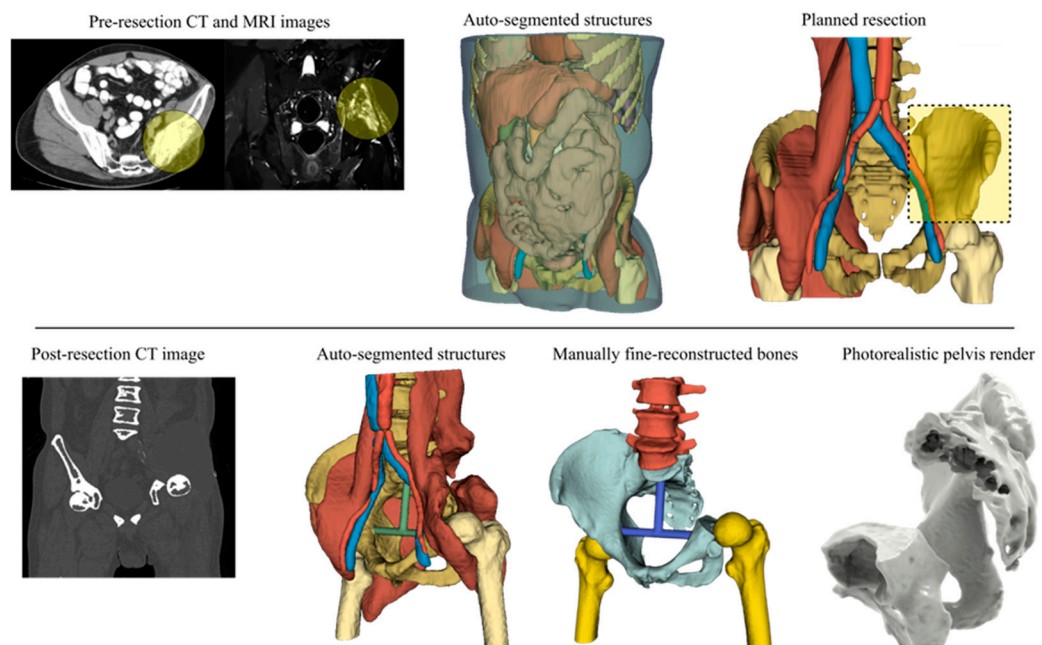

**Figure 4.** Pre- and post-resected 3D reconstructed anatomical models.

### *2.2. Step 2—Medical Image Acquisition*

The patient's anatomy before and after resection was CT-scanned with a Somatom Definition Flash (Siemens Healthcare GmbH, Erlangen, Germany) with a slice thickness of

0.75 mm. The image data in both cases were anonymized before being shared with the PoC unit for segmentation and the generation of the anatomical 3D models.

*2.3. Steps 3 and 4—Image Data Processing and Anatomy Visualization*

For both pre- and post-resected CT scans, the free and open-source DICOM processing software 3D Slicer [55] was selected to generate the 3D anatomical structures and bony models by applying both manual and automatic tools, including trained algorithms for auto-segmentation (Figure 4). The structures were auto-segmented and generated in 3 min, while finer results of the post-resected pelvis required 170 min using manual tools to export the entire CT-scanned bony compartment. The pelvis was isolated, exported into a 33 MB .stl file, and re-meshed into a 20 MB .stl file using the free and open-source software Meshlab (version 5.2.2) [56].

The digital 3D anatomy was visualized in static photorealistic views and interactive 3D models (Figure 4) generated by KeyShot (Luxion Inc., Costa Mesa, CA, USA), and these, along with the .stl file itself, were transmitted to the surgeon. A 3D anatomical model was also prepared to be (semi)immersively viewed in (Figure 5) (a) virtual reality using the Oculus Rift S (Lenovo Group Limited, Beijing, China) headset running the Diffuse (now Specto Medical) (Specto Medical AG, Basel, Switzerland) and Medicalholodeck VR (Medicalholodeck AG, Zurich, Switzerland) software and (b) augmented reality using an Apple iPad 2021 Pro to augment the post-resected mesh model using both Augment AR Viewer (Augment SAS, Paris, France) and Medicalholodeck's Medical Imaging XR applications.

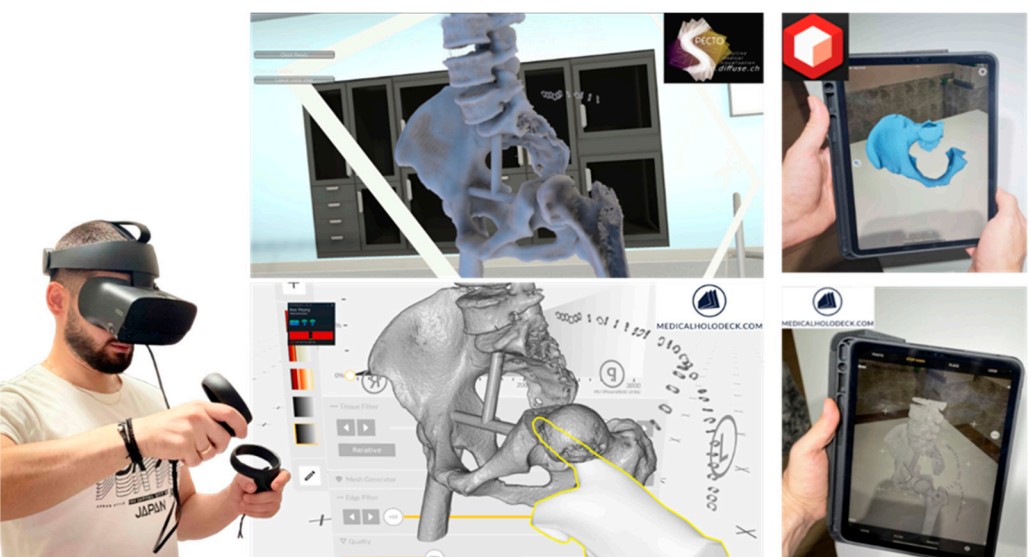

**Figure 5.** User immersed in virtual reality interacting with the post-resected pelvis in Specto and Medicalholodeck VR (**left**) and the post-resected model in AR Viewer and DICOM data in Medical Imaging XR (**right**).

A 248 × 231 × 168 mm pelvis was physically realized in a true-scale physical replica as a single monolithic piece (Figure 6) using the Creality (Shenzhen Creality 3D Technology Co., Ltd., Shenzhen, China) CR-10S 3D printer and ColorFabb's (Colorfabb B.V., Belfeld, the Netherlands) ivory-color PLA/PHA feedstock material. The free and open-source PrusaSlicer 2.4.2 was selected to plan the 3D printing process. In total, 663 g (EUR 23.2) of material was consumed in 3 days and 13 h. The pre-processing and post-processing took 15 and 20 min, respectively. The end product was more than acceptable, per se, in terms of surface finish and dimensional precision; measurements using a calibrated digital caliper revealed deviations of less than 0.3 mm.

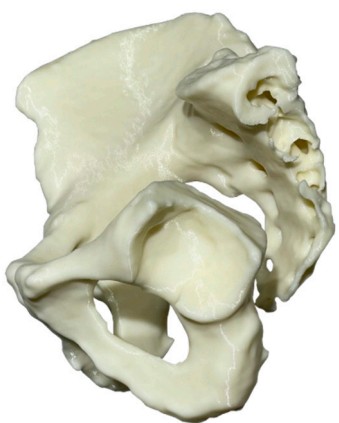

**Figure 6.** Post-resected 3D-printed pelvic model.

### 2.4. Steps 5 and 6—Rehabilitation Strategy and Task Clarification

The ultimate goal was to preserve the patient's limb and retain favorable functions in the long term by re-establishing the continuity of the pelvic girdle to prevent subsequent collapses and deformations in the residual bony tissue under load-bearing activities. As pelvic endo-prostheses are fraught with difficulties, the developed solution had to mitigate as much complexity in the operating room as possible. By examining the patient's post-resected DICOM data, the 3D digital model, the virtual model, and the physical 3D-printed model, the surgeon was able to outline the need for a non-bulky, low-profile endo-prosthesis conforming to the inner form of the resected bone tissue, and together with the PoC unit, they set the rehabilitation strategy and clarified the task by defining the specifications and requirements of the solution. Certain product design specification elements were utilized by considering the application that the endo-prosthesis should adhere to. The specifications were filled collaboratively within a 1 h meeting session, and a 24 h window for amendments was agreed upon.

### 2.5. Steps 7 and 8—Conceptual Design and Evaluation

Working principles and ideas were explored by searching a knowledge database with inputs sourced from natural systems, industrial sectors, the orthopedic device market, and the literature. Matching products, forms, functions, and principles were captured as hypodigms for inspiration and mimesis to turn working ideas after analysis and synthesis into working concepts, adapting the concepts to the specification list. A mind map of concepts for each function was created by brainstorming, and a 24 h window for amendments was agreed upon. Under SWOT analysis, from which threats were considered for risk assessment, the generated concepts were evaluated collaboratively based on preset constraints and weighted (from 1 to 5) criteria. The deliverables were hand sketches (30 min required) and a corresponding tangible red modeling clay replica of the concept achieving the best score (40 min required), and 3D marking in computer modeling software and virtual reality were also explored (Figure 7).

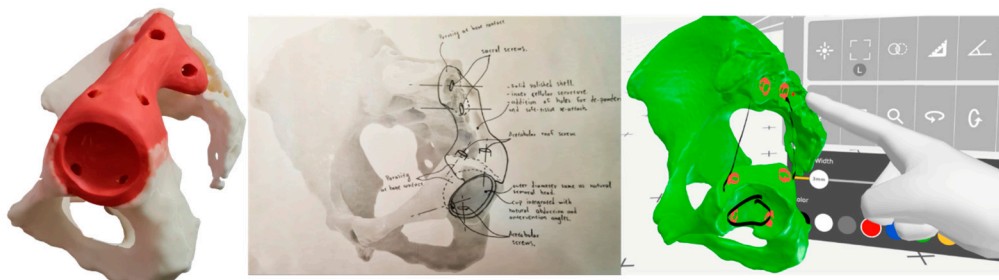

**Figure 7.** Physical clay modeling, hand sketching, and virtual reality sketching of the concept.

*2.6. Steps 9, 10, and 11—Embodiment and Detailed Design*

Then, 3D pre-operative planning and 3D modeling were conducted using the Space-claim 2022 R1 (Ansys Inc., Canonsburg, PA, USA), the cloud-based Fusion 360 (Autodesk Inc., San Fransisco, CA, USA), and Rhinoceros 7 (Robert McNeel and Associates, Seattle, WA, USA) software.

As patient-specific metrics have been suggested to be considered for cup placement instead of the typical "safe-zone" values reported in the literature decades ago [57], the anteversion (13.3°) and abduction angles (39.8°) were measured in 3D. A suitable off-the-shelf Ti alloy femoral stem with Ti coating that matched our needs, maintained the same symmetry, and eliminated leg-length discrepancy was selected along with a matching 22 mm diameter CoCrMo femoral head in combination with a UHMWPE 22/38 mm mobile liner within a stainless steel AISI 316L dual mobility 38/44 mm acetabular cup, which was cemented to the endo-prosthesis's 46/54 mm acetabular shell. The new left center of rotation had an 8.80 mm medial offset from the natural femoral head center and a 3 mm anterior offset from the right femoral head.

Six cancellous self-tapping Ti bone screws were positioned and oriented as per the bone morphology, surrounding soft tissue, and bone quality. Two 4 mm diameter cancellous screws were inserted 25 mm into the pubis and into the ischium as per the quadrant system described by Wasielewski et al. [58], two 6.5 mm diameter cancellous screws were inserted 30 mm from the acetabulum rooftop, and two 6.5 mm diameter cancellous screws were inserted 50 mm within the sacrum; most males and females can accommodate up to 8 mm diameter screws in S1 and S2 [59].

As hemispherical cups exhibit better stability with fewer complications compared with elliptic cups [60], a 46/54 mm diameter hemispherical acetabular shell was modeled with a spheroid-textured inner surface cup to facilitate better anchorage of the bone cement that fixed the cup in place. A solid endo-prosthesis was modeled and united with the acetabular shell into a single object. Unnecessary material was removed, and de-powdering holes were added for possible soft tissue suturing. Housings for all bone screw heads were added, and all features were blended into a smooth geometric result with no sharp edges.

Ancillary patient-specific instruments were also developed (Figure 8): (a) an acetabulum rooftop and sacrum side flat cutting jig; (b) an acetabulum reaming check tool used during reaming to guarantee a perfect fit, as the osseointegration of suitably interconnected porous implants depends on the minimization of micro-gaps, a reduction in micro-motions, and reaming for good vascularization and good bleeding [61]; (c) a femoral head cutting jig; and (d) bimodal bone screws, pilot hole drilling, and a trial endo-prosthesis replica.

By ignoring (a) the effects of all surrounding muscles, ligaments, and cartilage; (b) the threads of the bone screws and their tightening and reaction forces; and (c) the anisotropic characteristic of 3D-printed products, a mechanical static stress analysis was conducted in Fusion 360 (Nastran solver), as shown in Figure 9. Ti-6Al-4V material properties were assigned to the bone screws and endo-prosthesis, whereas the pelvic bones were assumed to behave as a single homogeneous, linear isotropic, elastic material with an averaged modulus of elasticity of $E = 7$ GPa and a Poisson's ratio of $\nu = 0.3$ [62,63]. The sacrum was fixed by constraining the top surfaces in all degrees of freedom, and a force four times the relevant body weight to simulate walking and stair-descending activities [64] (rounded at 3000 N) was applied individually and simultaneously at both hip joints. The bodies were volume-meshed with tetrahedral elements, and the simulation generated Von Misses stresses and displacement results. The stress distribution and displacement of the pelvic bones, endo-prosthesis, and bone screws differed considerably in each scenario with the peak stresses being considerably (16 times) lower (<60 MPa) than the yield stress of the endo-prosthesis and bone screws. As the relationship between load and stress is linear, a force eight times the body weight (to simulate the extreme event of stumbling [65]) could be easily borne by the endo-prosthesis and the bone screw materials.

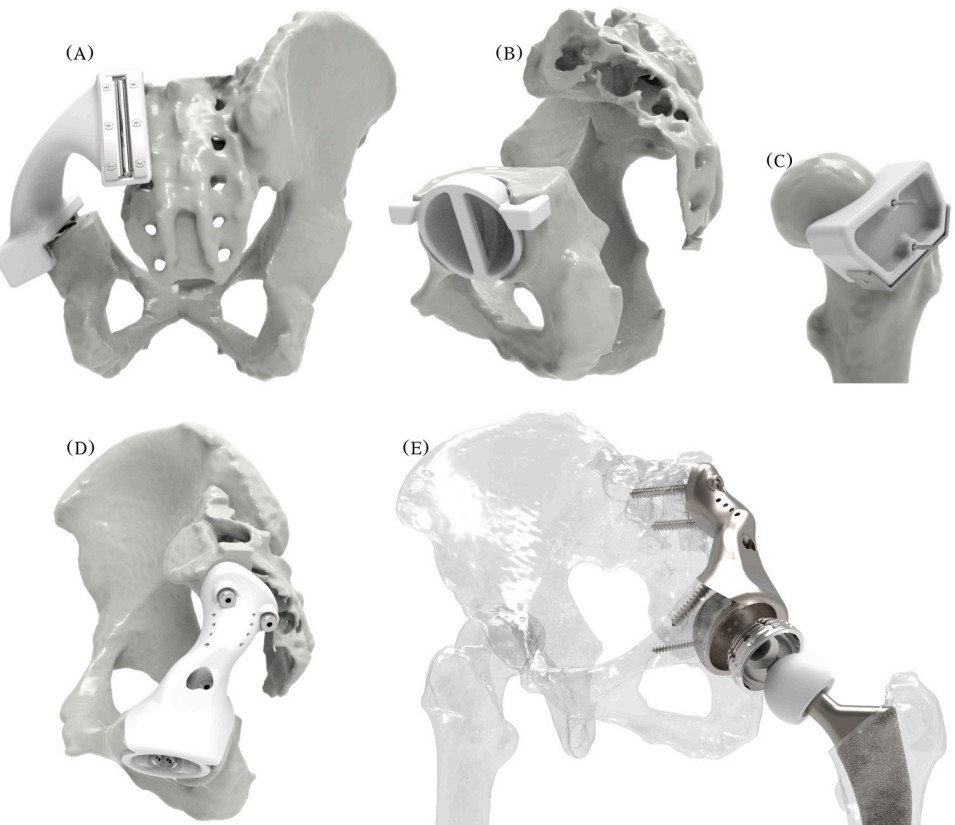

**Figure 8.** Photo-realistic models of the (**A**) cutting jig, (**B**) reaming check tool, (**C**) femoral head cutting jig, (**D**) drilling jig–trial endo-prosthesis instrument, and (**E**) endo-prosthesis and the total hip arthroplasty components.

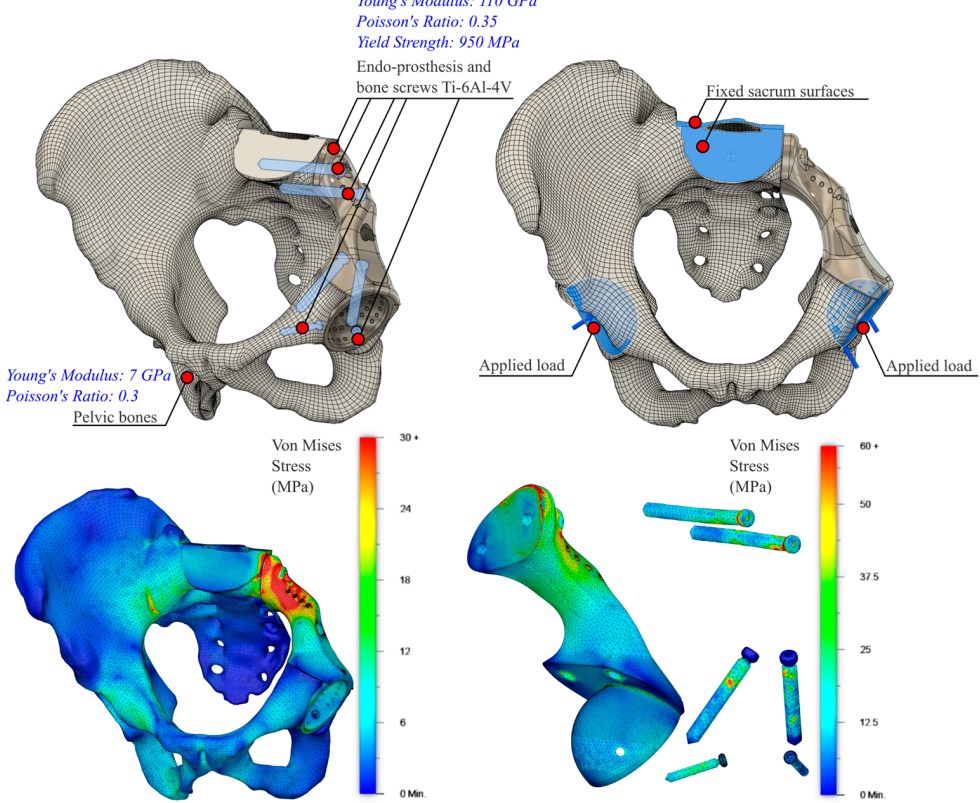

**Figure 9.** Mechanical static stress simulation and results for the combined hip-loading scenario.

The higher modulus of elasticity and the solid endo-prosthesis form result in a much stiffer endo-prosthesis compared with the host pelvic bone. Although topology optimization or generative design could have been employed to lighten the initially fully solid 416 g endo-prosthesis down to 292 g or 179 g, respectively (Figure 10) via stress-driven mass reduction, instead, we decided to introduce cellular architecture to the endo-prosthesis and keep a thin solid outer shell.

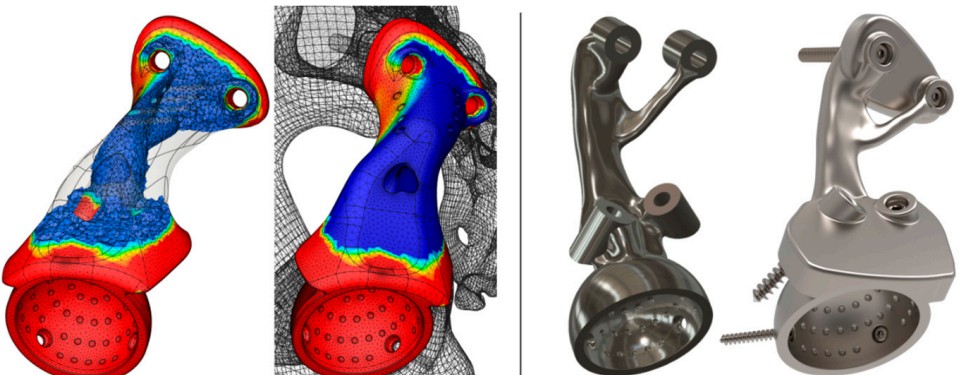

**Figure 10.** Topology-optimized (**left**) and generatively designed endo-prosthesis (**right**).

Toward the goal of "biologizing" the internal form and external surfaces that touch the host bone and obtaining optimal bimodal performance (bone ingrowth and stiffness reduction) within a suitable design space, it was concluded that the spongy complex of the natural bone can be mimicked by a functionally graded stochastic Voronoi foam pattern that can be assigned both internally and at bone-contact surfaces by considering (a) an optimal pore size of 0.3–0.6 mm for good cell colonization, vascularization, mechanical integrity, and permeability [66,67]; (b) manufacturability (laser beam powder bed fusion); and (c) the histomorphometric [27] properties of the trabecular bone at the iliac crest, i.e., 84% porosity, 0.15 mm feature thickness, and 0.75 mm pore size. Using nTopology 4.12.2 (nTopology Inc., New York, NY, USA), the endo-prosthesis was architected selectively and topologically with a gradient and conforming 3D stochastic Voronoi pattern. The total mass of the resultant optimized endo-prosthesis meta-(bio)material totaled 256 g, with 151 g being the scaffold structures and 105 g being the hollowed body with 1–1.5 mm wall thickness. The characteristics of the optimized biomimetic result are shown in Figure 11.

The generated stochastic Voronoi structure was assumed to be isotropic and homogeneous in terms of its elastic properties along any loading direction because of its randomized cell orientation [68]. Voronoi-based foams with open cells and high porosities are considered ideal bending-dominated structures, with a resultant modulus of elasticity calculated with the following formula [68,69]:

$$E\_c = E\_s * \rho \char`^ 2 \tag{1}$$

where E_c and E_s are the modulus of elasticity of the cellular and fully dense solids, respectively, and $\rho$ is the relative density of the cellular solid, which is connected to the porosity ($\varphi$) of the cellular solid by the formula

$$\varphi = (1 - \rho) * 100 \tag{2}$$

In this case, as per Equations (1) and (2), the resultant modulus is E = 17.6 GPa, which is close to the modulus of elasticity of cortical bone [70]. To assess the mechanical performance of the final optimized form, only the hollowed body was used to carry out a similar mechanical static stress analysis as before. The hollowed body withstood the same loads as exerted previously on the fully solid endo-prosthesis, but in this case, under greater (three-fold) stresses and displacements. It was orthological to assume that introducing the

stochastic Voronoi infill would not compromise the mechanical performance of the solid shell and would not lead to mechanical failure, as plastic deformation is unlikely since the cellular structure is stiffened by the solid shell, though this claim should have been verified by suitable extra simulations.

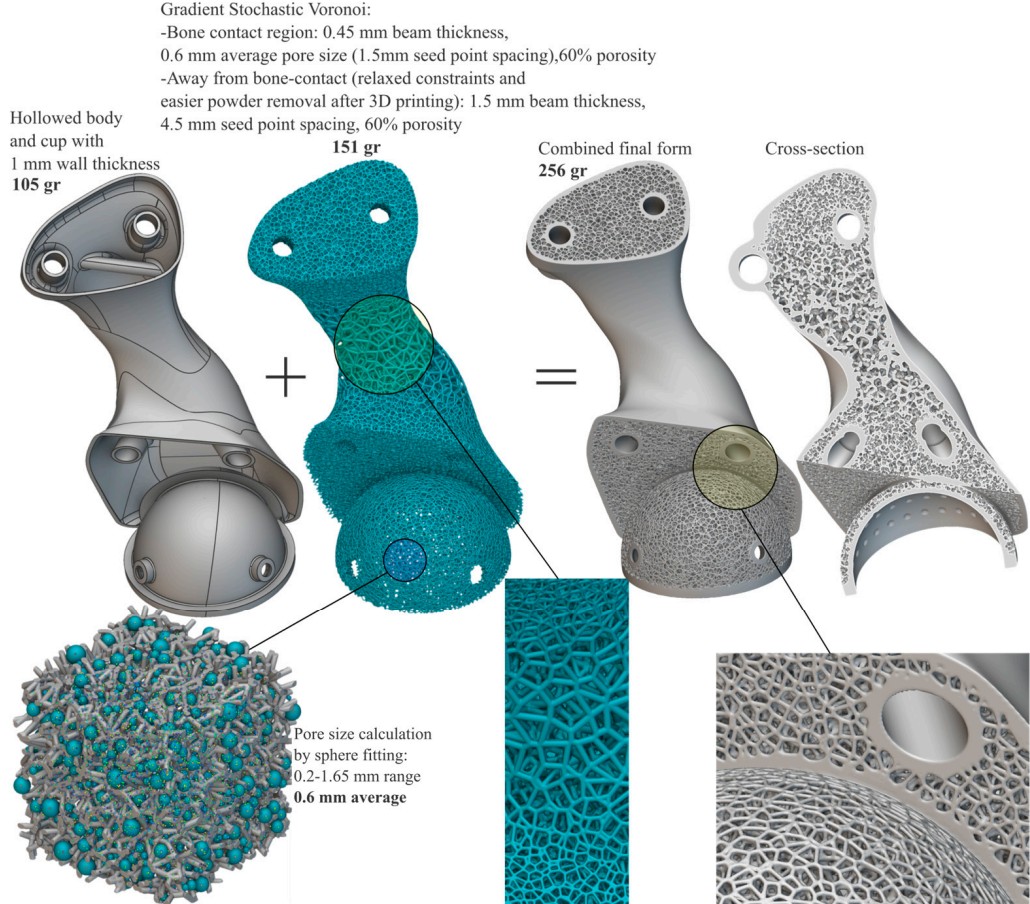

**Figure 11.** Mass and characteristics of the gradient stochastic Voronoi-optimized endo-prosthesis.

The total time to complete the embodiment and the detailed design was 3 days.

### 2.7. Step 12—Prototyping

Digital, virtual, and physical visualization was conducted for verification of all components of the solution (Figure 12). An anatomical model with a reamed acetabulum and flat-cut surfaces; a femoral head; a femoral head with a femoral stem and head inserted; a cemented cup; and prototypes of the cutting jig, reaming check tool, femoral head cutting jig, drilling jig–trial endo-prosthesis instrument, and endo-prosthesis were 3D-printed in three sessions using the Creality CR-10s 3D printer in 3 days and 13 h, 2 days and 4 h, and 22 h, consuming 663 g (ivory, EUR 23.2), 370 g (gray, EUR 13), and 280 g (black, EUR 10) ColorFabb PLA/PHA material, respectively. The total pre-processing and post-processing took 20 and 30 min, respectively. To better illustrate the details of the optimized endo-prosthesis, the Elegoo (Elegoo Inc., Shenzhen, China) Saturn S vat photopolymerization 3D printer and a clear plant-based photocurable resin were used. Chitubox Basic 1.9.4 (Chuangbide Technologies Co., Ltd., Shenzhen, China) was used to plan the process, which took 6 h, consuming 55 mL (EUR 3) of resin.

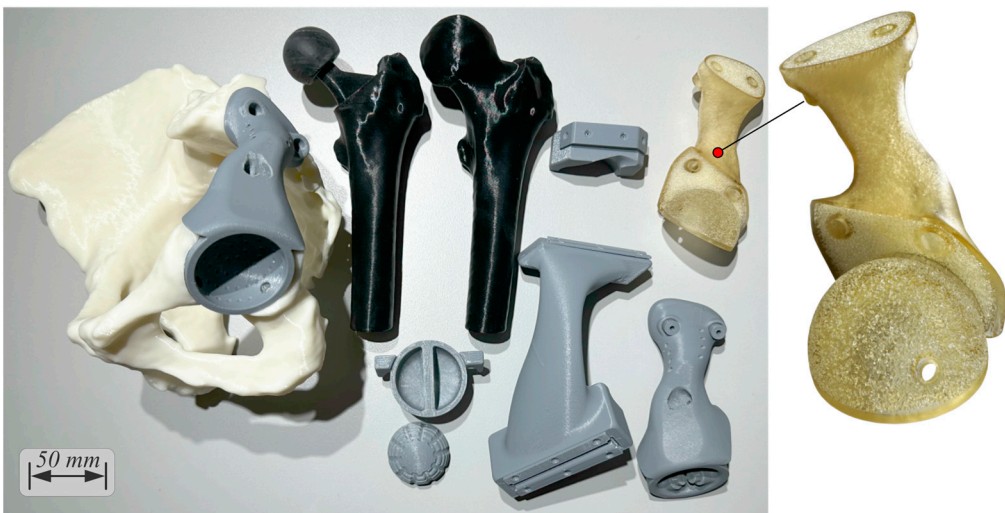

**Figure 12.** The 3D-printed prototype parts of the solution.

*2.8. Steps 13, 14, and 15—Final Meeting, Manufacturing, and Implantation*

All characteristics of the solution were documented, a final meeting with all required parties was arranged for final approval as per the applicable legal and standard procedures, and labels (unique numbers) were embossed on the surface of each unique part. The off-the-shelf components (total hip arthroplasty, bone screws, K-wires) were readily available on the market for purchase, whereas all custom parts were sent to a contract manufacturer; i.e.,

1. The cutting jig and drilling jig-trial endo-prosthesis instrument with all metal inserts, the reaming check tool, and the femoral head cutting jig were ordered to be 3D-printed out of surgical-grade PA12 with a suitable powder-bed fusion technology (laser sintering). The total cost was EUR 380 for 6 working days and delivery.
2. The endo-prosthesis was ordered to be 3D-printed out of surgical-grade Ti-6Al-4V with a suitable powder-bed fusion technology (laser melting), followed by glass bead blasting for satin surface finish. The total cost was EUR 1800 for 6 working days and delivery.

Once received and before application, all parts were carefully inspected, cleaned, and sterilized to prevent infection by applying the most suitable method [71], as erroneous or inadequate cleaning, post-processing, sterilization, and packaging can have serious performance and safety consequences.

## 3. Results and Discussion

The COVID-19 pandemic catalyzed the need to adopt a new model that embraces digital health tools [72], including xR devices, 3D design software, and 3D printing. As the weaknesses of modern healthcare systems were revealed, and their capacity, supply, and workforce were challenged, 3D printing proved to have suitable characteristics and potential, providing a glimpse into the future of healthcare by enabling the decentralized (localized) manufacturing of personal protective equipment and other critical parts at the PoC [73] despite some safety concerns and challenges [74]. Similarly, patient-specific medical devices can be designed and/or manufactured just in time at the PoC at affordable costs and with short lead times, provided that suitable equipment and infrastructure exist [30].

The operation of a PoC unit lies at the intersection of healthcare providers, regulatory groups, engineers, and manufacturers who contribute to the common goal of providing high-quality and safe solutions to each unique patient. In the quest to establish PoC units, non-technical challenges (regulatory compliance, lack of directives, standards and quality management systems, need for skilled personnel, intellectual property, and digital security

issues) and technical limitations still hold 3D technologies from springboarding and should be addressed [8]. PoC units must establish their best practices by implementing quality management systems while also conforming to applicable regulations and standards. In the USA and the EU, custom-made emergency devices can be exempted from regulatory control by allowing authorized healthcare professionals to provide design input and take responsibility for the safety and performance of customized medical devices. However, because existing regulations and standards oversee the way medical devices are manufactured, 3D technologies need further guidance to establish quality control measures as numerous parameters might lead to a low-quality 3D-printed product, i.e., data preparation variables, design features, feedstock materials, process and 3D printer parameters, personnel, documentation, infrastructure, and post-processing. Therefore, appropriate quality management systems, including strategies and processes for addressing errors and improving consistency, are important to controlling and assuring quality and safety. Otherwise, the uncontrolled rapid growth of 3D technologies and the increasing complexity might put patients at increased risk of potential errors, decreased quality, and sub-optimal performance for the solution. Ultimately, the evolution and synthesis of regulations, quality management systems, and accessibility to 3D technologies will define the maturation of PoC patient-specific device development. As necessity drives invention and innovation, once the readiness of 3D technologies stops remaining under question, data protection issues are addressed, and costs in tandem with medical device development time are further reduced, their PoC incorporation will accelerate healthcare into intelligent formats capable of unlocking mass customization with high-quality end results and increased patient satisfaction.

A plethora of trade-offs can be made for PoC units in terms of software and hardware equipment; infrastructure and layout organizational structure; and personnel. PoC fabrications of end parts could eliminate reliance on external partners, potentially reducing costs; however, it is recommended that in-house 3D designs and 3D printing should be commenced only for prototyping and outsourcing end parts. Only if sufficient demand exists is it orthological to increase capacity and implement in situ final manufacturing. Investments should be rationalized and driven by clinical needs so that collaboratively developed solutions are clinically and scientifically relevant and ultimately beneficial for each unique patient.

The developed patient-specific solution was successfully prototyped at the PoC within an acceptable clinical time and cost frame. The end polymer and metallic parts that required high-cost industrial-grade systems were outsourced to external partners, but 3D-printing polymer anatomical models, surgical guides, and implant prototyping, along with xReality systems for pre-operative planning and intra-operative (non-navigation) purposes, can be implemented at relatively low initial capital investments and operating and maintenance costs. The most time-consuming tasks were found to be the 3D printing of anatomical models and all prototype parts of the solution, but since much faster, low-budget 3D printers are currently available in the market, the time can be considerably reduced.

The custom meta-(bio)material-by-design endo-prosthesis developed for pelvic reconstruction after tumor resection featured interconnected porous architectures with gradient pore size and beam thickness. The areas away from the bone had a larger pore size and thicker beams to facilitate better manufacturability and the removal of trapped and un-melted powder particles after laser-based powder-bed fusion 3D printing, while the bone-contact regions had values as recommended in the literature, e.g., ~60–70% porosity with ~0.6 mm pore size [66,67]. Numerous 3D-printed implantable devices featuring cellular architectures have been researched, commercialized, and cleared by the FDA [12,13], and postoperative follow-ups of such porous morphologies have been proven to have good short- and mid-term outcomes [3,75–77], including hemi-pelvic endo-prostheses that have shown good oncological and functional outcomes with stable fixation and good osseointegration without severe complications [78–84].

With 3D printing, standard, modular, and patient-specific parts can be materialized that cannot or are extremely difficult to make using traditional methodologies. On the other hand, 3D printing is not a panacea, as it is prone to challenges when biomimetic porous structures and fine details are involved. Hence, appropriate design rules for ensuring manufacturability using the selected 3D printing technology must be adopted to avoid differences between as-designed and as-3D-printed products [85]. Appropriate methods should be implemented for the powder-bed fusion 3D printing of metals [86] as multi-scale, multi-physics phenomena occur that might lead to structural defects and implications if sub-optimal processing parameters are selected [27,87,88]. For example, the de-powdering of un-melted and trapped powder particles after the completion of laser-based powder-bed fusion 3D printing is critical, as there is a risk of particles entering the patient's body; air-jetting followed by dry ultrasonic vibration and/or chemical etching is important [89]. In addition, if the surface finish, dimensional accuracy, and fatigue properties have to be improved, post-processing methods such as machining and heat treatments can be applied.

Toward this direction, algorithmically driven and machine learning methodologies powered by trained datasets and sourced from the libraries of previous cases with an emphasis on design manufacturing process optimization for the highest mechanical and biological performance are highly sought in order to minimize errors and defects [90]. As image data acquisition and processing and design and manufacturing techniques are streamlined in tandem with artificial intelligence methods for decision-making and automating processes, the range of applications will widen, and ultimately the reliance on large, centralized manufacturing plants and the maintenance of expensive supply networks might be reduced.

This case study had several limitations that will be researched in the future:

1. To better define the endo-prosthesis form and its fixation points, directions, and lengths, multi-modal CT, MRI, magnetic resonance neurography (MRN), and computed tomography angiography (CTA) imaging are required to account for neurovascular tissue [91,92].

2. Extra-mechanical simulations considering anisotropic and inhomogeneous bone tissue properties [93,94] and surrounding soft tissue; the threads of bone screws and their tightening reaction forces; and dynamic fatigue analyses should be executed by considering more loading scenarios such as direct impacts caused by sideways falls.

3. Regarding the mechanical performance of the spongy architectures, alternative functionally graded structures could be explored and simulated while also investigating local stress concentrations, contact phenomena, and deformation levels on the nodes and beams to ensure that these can withstand the exerted loads without failure. Ideally, cellular structures should be generated by simulation-driven multi-objective and multi-scale tools that account for spatial variations in parameters across parts, per se.

4. Research on coating substances that can be applied to the porous surface for bacterial-fighting and osseointegration-boosting activities.

5. More case studies should enrich the portfolio, and follow-ups must be reported.

Future research can be extended toward scaffold-based or scaffold-free bioprinting for bone grafting; multi-material 3D printing of hard-soft composites; 4D (bio)printing of stimuli-responsive materials; drug delivery systems that elute under control; and "instrumented" endo-prostheses capable of sensing, measuring, and reporting real-time physical quantities.

## 4. Conclusions

Healthcare has traditionally been a non-engineering design and non-manufacturing sector, as neither design nor manufacturing usually takes place in situ. The entrance of 3D design and manufacturing technologies is transforming healthcare facilities into decentralized design and/or manufacturing sites. PoC units aiming at medical device development should be established accordingly to facilitate the symbiosis of healthcare professionals with skilled (bio)mechanical engineers to develop patient-specific devices at the PoC within acceptable lead times and costs.

Three-dimensional technologies are newly inserted tools into the modern digital cosmos of individualized healthcare delivery and introduce advanced abilities for delivering medical devices at the PoC with greater customization, innovation in design, cost-effectiveness, and high quality. Although related pitfalls are known to engineers, an increased awareness is suggested to explore and exploit the full potential of the 3D design–3D printing–xReality triad in realizing medical devices at the PoC under suitable quality management systems and as per the applicable regulations and standards.

**Author Contributions:** Conceptualization, I.I.M. and I.G.T.; methodology, I.I.M., I.G.T. and V.A.K.; software, I.I.M.; validation, V.A.K., O.D.S. and P.J.P.; formal analysis, V.A.K. and O.D.S.; investigation, I.I.M. and I.G.T.; resources, I.I.M. and P.J.P.; writing—original draft preparation, I.I.M. and I.G.T.; writing—review and editing, I.I.M., O.D.S. and P.J.P.; visualization, I.I.M.; supervision, V.A.K., O.D.S., and P.J.P.; project administration, I.I.M. and P.J.P. All authors have read and agreed to the published version of the manuscript.

**Funding:** This research received no external funding.

**Institutional Review Board Statement:** Not applicable.

**Informed Consent Statement:** Not applicable.

**Data Availability Statement:** No data are reported.

**Conflicts of Interest:** The authors declare no conflict of interest.

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
