# Peer review of "Point-of-Care Orthopedic Oncology Device Development"

_curroncol, doi:10.3390/curroncol31010014_

Round 1
Reviewer 1 Report
Comments and Suggestions for Authors
The manuscript explores leveraging 3D design, 3D printing, and virtual/augmented reality for personalized medical solutions, particularly in orthopedic devices. It shows a case study on pelvic reconstruction post-osteosarcoma resection to demonstrate the viability of an integrated workflow. Barriers and potential for these technologies are discussed, emphasizing the need for collaboration between orthopedists and engineers. Ultimately, the article advocates for heightened awareness to harness the full potential of these technologies in delivering highly tailored and cost-effective medical devices at the Point-of-Care.
This study aligns well with the journal's scope and presents interesting insights. However, the introduction lacks sufficient background development on the topic. To strengthen the study, establishing in-house capabilities for producing the necessary polymer and metallic parts for patient-specific solutions could diminish reliance on external partners, potentially reducing costs. Moreover, advancing research into imaging techniques, mechanical simulations, and alternative structures could significantly enhance the precision and efficacy of the solutions. Implementing algorithm-driven methodologies and leveraging datasets for design optimization may further elevate the mechanical and biological performance of these structures.
Author Response
Dear Reviewer,
Thank you very much for taking the time to review this manuscript. Please find the detailed responses below:
- The abstract was reduced in word count.
- The manuscript was re-formatted, language-checked and sub-headings were added.
- The Point-of-Care term is now used in its abbreviated form (PoC).
- The introduction was enriched with material, and the scope was highlighted. An extra figure was added.
- Figure 10 (now Figure 11) caption was corrected, the graphics were enriched with extra information and the font size was enlarged.
- The extra word "For" in steps 3 and 4 was un-bolded.
- The extra word "3D” in Steps 9, 10 and 11 - Embodiment and Detailed Design was un-bolded. New text was added at the end. Also, it was added that the anisotropic characteristics of 3d printed products were not considered when performing the static stress analysis of the hip structure.
- A scale bar was added in Figure 11 (now Figure 12).
- New text was added in the first paragraph of Steps 13, 14 and 15 - Final meeting, manufacturing and implantation.
- The results and discussion section was enlarged and references were added.
- Conclusions were slightly altered.
- The total word count from Abstract to end of Conclusions is now 5425 words.
- The institutional e-mail addresses were inserted.
- A typo was corrected in the institution name.
- The correct reference style format was applied.
The need to advance research into imaging techniques, mechanical simulations, and alternative structures was highlighted. Also, algorithm-driven and machine-learning methodologies by leveraging datasets for design optimization and automating processes were mentioned.
Reviewer 2 Report
Comments and Suggestions for Authors
General comment:
the paper proposes the use of 3D design, 3d printing, and AR/VR technologies to create just-in-time patient specific products. the workflow was described clearly. it can be further improved by addressing the following concerns.
Specific comments:
1. one advantage of 3d printing is the ability to print hard and soft materials to achieve tissue mimicking effect, which is crucial in preoperational planning.
a. Goh, G. D., Sing, S. L., Lim, Y. F., Thong, J. L. J., Peh, Z. K., Mogali, S. R., & Yeong, W. Y. (2021). Machine learning for 3D printed multi-materials tissue-mimicking anatomical models. Materials & Design, 211, 110125.
b. Bezek, L. B., Cauchi, M. P., De Vita, R., Foerst, J. R., & Williams, C. B. (2020). 3D printing tissue-mimicking materials for realistic transseptal puncture models. Journal of the Mechanical Behavior of Biomedical Materials, 110, 103971.
2. in steps 3 and 4. an extra word "For" was bolded.
3. in steps 9 and 10 and 11. an extra word "3D" was bolded.
4. figure 10, font size is too small to read clearly.
5. add a scale bar in figure 11
6. was anisotropic characteristic of the 3d printed material considered when performing the simulation of the hip structure?
Author Response
Dear Reviewer,
Thank you very much for taking the time to review this manuscript. Please find the responses below:
- The abstract was reduced in word count.
- The manuscript was re-formatted, language-checked and sub-headings were added.
- The Point-of-Care term is now used in its abbreviated form (PoC).
- The introduction was enriched with material, and the scope was highlighted. An extra figure was added.
- Figure 10 (now Figure 11) caption was corrected, the graphics were enriched with extra information and the font size was enlarged.
- The extra word "For" in steps 3 and 4 was un-bolded.
- The extra word "3D” in Steps 9, 10 and 11 - Embodiment and Detailed Design was un-bolded. New text was added at the end. Also, it was added that the anisotropic characteristics of 3d printed products were not considered when performing the static stress analysis of the hip structure.
- A scale bar was added in Figure 11 (now Figure 12).
- New text was added in the first paragraph of Steps 13, 14 and 15 - Final meeting, manufacturing and implantation.
- The results and discussion section was enlarged and references were added.
- Conclusions were slightly altered.
- The total word count from Abstract to end of Conclusions is now 5425 words.
- The institutional e-mail addresses were inserted.
- A typo was corrected in the institution name.
- The correct reference style format was applied.
- The suggested papers were referenced.
Reviewer 3 Report
Comments and Suggestions for Authors
The article Point-of-Care Orthopaedic Oncology Devices Development, presents an intriguing exploration of the integration of 3D design, 3D printing, and virtual/augmented reality technologies in the development of patient-specific orthopaedic devices. The background sets a promising premise, highlighting the significance of meta-(bio)materials in the context of Point-of-Care applications. The objective to evaluate the feasibility and potential of these technologies in creating personalized medical devices is both ambitious and timely, considering the evolving landscape of personalized medicine. The methodological choice of a case study focusing on pelvic reconstruction post-osteosarcoma resection is commendable for its specificity and relevance. The detailed explanation of the inter-epistemic workflow, involving a collaboration between orthopaedists and (bio)mechanical engineers, is thorough and well-articulated. However, the study could benefit from a larger sample size or multiple case studies to strengthen the generalizability of the findings. The discussion on the potential of 3D technologies and the barriers to their adoption is insightful and well-balanced. The article effectively highlights the symbiotic relationship between various disciplines in maximizing the benefits of these technologies at the Point-of-Care. However, the discussion could be enriched by providing more empirical data or statistical analysis to support the claims. The exploration of image-based diagnosis and treatment, along with the practical aspects of using 3D models and xReality environments, is innovative and sheds light on the practical applications of these technologies. The conclusion advocating for increased awareness to fully exploit the 3D design-3D printing-xReality triad is persuasive and well-founded. The emphasis on customization, cost-effectiveness, and quality in the development of medical devices is particularly relevant in the current healthcare environment. Nevertheless, the article could further elaborate on the long-term implications and sustainability of these technologies in clinical practice. The article is generally interesting but has several errors and needs to be refined. My detailed comments are included below:
Editing and text preparation
The paper contains numerous editorial errors and needs a thorough formatting check. If all authors are from the same unit there is no need to give a separate link to each. Please check o correct the formatting according to the journal's guidelines.
Abstract
The abstract is too long, please rebuild it leaving only the most important information about the work.
Introduction
The introduction is too short and does not sufficiently mark the essence of the problem undertaken in the work. I ask you to supplement the introduction with detailed information on the potential for the application of rapid prototyping and 3D printing in orthopedic surgery, and then clearly indicate the purpose of the work and the essence of the results obtained. Please add the necessary and up-to-date literature. You will find useful information in the papers:
DOI 10.3390/cells12060859
DOI 10.3390/ma15144731
DOI 10.3390/bioengineering10060644
Materials and Methods
The materials and methods adopted in the paper are described correctly and I have no comments on this part.
Results and Discussion
The results and discussion chapter is quite short. Please expand the discussion and analyze more work by other authors with necessary additions to the literature. The paper contains only 30 references which is far too small a number especially with a topic so widely described.
In conclusion, this article is a valuable read for professionals in the fields of orthopaedic oncology, biomedical engineering, and healthcare technology. It successfully paves the way for further research and development in the realm of personalized medical devices at the Point-of-Care, marking a step forward in the intersection of technology and patient-centered care. After making the appropriate corrections and additions in my opinion, it can be accepted.
Comments on the Quality of English Language
Minor editing of English language required
Author Response
Dear Reviewer,
Thank you very much for taking the time to review this manuscript. Please find the responses below:
- The abstract was reduced in word count.
- The manuscript was re-formatted, language-checked and sub-headings were added.
- The Point-of-Care term is now used in its abbreviated form (PoC).
- The introduction was enriched with material, and the scope was highlighted. An extra figure was added.
- Figure 10 (now Figure 11) caption was corrected, the graphics were enriched with extra information and the font size was enlarged.
- The extra word "For" in steps 3 and 4 was un-bolded.
- The extra word "3D” in Steps 9, 10 and 11 - Embodiment and Detailed Design was un-bolded. New text was added at the end. Also, it was added that the anisotropic characteristics of 3d printed products were not considered when performing the static stress analysis of the hip structure.
- A scale bar was added in Figure 11 (now Figure 12).
- New text was added in the first paragraph of Steps 13, 14 and 15 - Final meeting, manufacturing and implantation.
- The results and discussion section was enlarged and references were added.
- Conclusions were slightly altered.
- The total word count from Abstract to end of Conclusions is now 5425 words.
- The institutional e-mail addresses were inserted.
- A typo was corrected in the institution name.
- The correct reference style format was applied.
The first two of the suggested papers were referenced.
Round 2
Reviewer 1 Report
Comments and Suggestions for Authors
The authors have addressed the reviewers' comments well. The manuscript could be considered for publication.
Reviewer 2 Report
Comments and Suggestions for Authors
The replies are satisfactory. the quality of the manuscript has improved significantly in the revised version in terms of the content. but i can't comment on the formatting, the track changes make it hard to read.
Comments on the Quality of English Language
NA